

# Three-body correlations in mesonic-atom-like systems

Hajime Moriya[1]⋆, Wataru Horiuchi[1] and Jean-Marc Richard[2]

**1** Department of Physics, Hokkaido University, Sapporo 060-0810, Japan
**2** Institut de Physique des 2 Infinis de Lyon, Université de Lyon,
CNRS-IN2P3-UCBL, 4, rue Enrico Fermi, Villeurbanne, France

⋆ moriya@nucl.sci.hokudai.ac.jp

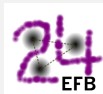

## Abstract

Three-body correlations in three-body exotic atoms are studied with simple models that consist of three bosons interacting through a superposition of long- and short-range potentials. We discuss the correlations among particles by comparing the energy shifts given by precise three-body calculations and by the Deser-Trueman formula, in which the long- and short-range contributions are factorized. By varying the coupling of the short-range potential, we evaluate the ranges of the strength where the two-body correlations dominate and where the three-body correlations cannot be neglected.

## 1 Introduction

A mesonic atom is a Coulomb bound system consisting of negatively-charged mesons surrounding a nucleus. Studying such systems gives access to the properties of the meson-baryon interaction at very low energy [1–7]. For example, this antikaon-nucleon ($\bar{K}N$) interaction is believed to be a strong short-range attraction as suggested if $\Lambda(1405)$ has a dominant $\bar{K}N$ structure [5,8]. The existence of bound kaonic nuclei is still under discussion and it is essential to improve our knowledge of the $\bar{K}N$ interaction [9]. The simplest atom, kaonic hydrogen, consists of an antikaon ($K^-$) and a proton ($p$). It was used to extract some information about the $\bar{K}N$ interaction [10,11]. A study of kaonic deuterium [6,10,11] gives interesting constraint on the isospin dependence of the $\bar{K}N$ interaction. Encouraged by these results, we investigate whether the physics of exotic atoms can be extended to three-body systems, without assuming that two of them form a nucleus. A preliminary study was made by one of the present authors (JMR) and C. Fayard [12], who considered a simple system of three identical bosons interacting via simple long- and short-range potentials. By varying the strength of the short-range term, they studied the level rearrangement of the spectrum, and the transition from atomic to nuclear states. They found that the contributions from long- and short-range potentials to the

energy shifts can be factorized within a certain range of the potential strength. Our aim is to extend this study, to consider more realistic cases treated in a more quantitative manner.

The paper is organized as follows: In the following sections, we introduce the models, the method to solve the three-body problem, and the method of determinant, to probe whether the energy shifts are given by a sum of products of long- and short-range terms.

## 2 Models

In this paper, two three-body models are employed.

### 2.1 Model I

The simplest model consists of three identical bosons. All interactions between two particles have long-range and short-range attraction parts. The Hamiltonian of this system is

$$H_{\text{I}} = \sum_{i=1}^{3} T_i - T_{cm} + \sum_{i>j=1}^{3} V_{ij}^{LR} + \lambda \sum_{i>j=1}^{3} V_{ij}^{SR}, \tag{1}$$

where $T_i$ ($i = 1, 2, 3$) is the kinetic energy of the $i$th particle and $T_{cm}$ is the kinetic energy of the center of mass, which is subtracted. All the physical constants including masses are set to 1. The long-range ($LR$) and short-range ($SR$) two-body potentials have only a central term. The strength of the short-range potential is varied through the parameter $\lambda$. We assume a regularized Coulomb for the long-range part and a Gaussian shape for the short-range potential. The explicit forms are

$$V_{ij}^{LR} = -\frac{\text{erf}(\mu_{LR} r_{ij})}{r_{ij}}, \tag{2}$$

$$V_{ij}^{SR} = -C_{SR} \mu_{SR}^3 \exp\left(-\mu_{SR}^2 r_{ij}^2\right), \tag{3}$$

where $r_{ij}$ denotes the distance between the $i$th and $j$th particles. The strength parameter $C_{SR}$ is tuned so that the the short-range potential alone supports a two-body bound state for $\lambda > 1$, i.e., $\lambda = 1$ is the coupling threshold for binding. The range parameters of both the short-range potential and the regularizing term of the long-range potential are set to $\mu_{LR} = \mu_{SR} = 30$, which is large compared to the inverse Bohr radius, so that the role of the long- and short-range interactions are well separated. Since all the long-range interactions are attractive, this model cannot be realized by Coulombic systems, it corresponds to a gravitational interaction.

### 2.2 Model II

Model II describes a case that is more realistic, or at least closer to the $ppK^-$ system. The first and second particles are identical bosons with a mass $m_1 = m_2 = 1$ and a positive charge $q_1 = q_2 = +1$, while the third particle, also spinless, has a mass $m_3 = 1/2$ and a charge $q_3 = -1$. The short-range potential is restricted to the interaction with the third particle, with $C_{SR}$ appropriately rescaled so that $\lambda = 1$ is the coupling threshold for a two-body system of masses $\{m_1, m_3\}$. The Hamiltonian of Model II is

$$H_{\text{II}} = \sum_{i=1}^{3} T_i - T_{cm} + \sum_{i>j=1}^{3} V_{ij}^{LR} + \lambda \sum_{i=1}^{2} V_{i3}^{SR}, \tag{4}$$

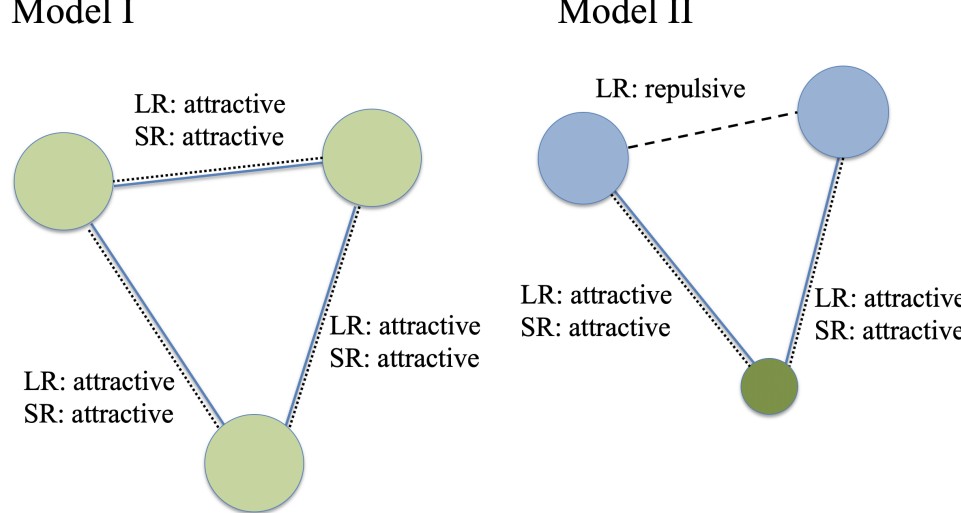

Figure 1: Schematic pictures of Model I and II employed in this paper. Solid lines represent the short-range attractive potentials, and dotted and dashed lines represent the long-range attractive and repulsive potentials, respectively.

with

$$V_{ij}^{LR} = q_i q_j \frac{\mathrm{erf}(\mu_{LR} r_{ij})}{r_{ij}}, \tag{5}$$

$$V_{ij}^{SR} = -C_{SR} \mu_{SR}^3 \exp\left(-\mu_{SR}^2 r_{ij}^2\right). \tag{6}$$

Model I and Model II are schematically summarized in Fig. 1.

## 3 Correlated Gaussian expansion

The three-body calculations are carried out by a well-known variational method, which is now briefly summarized. Let $x$ denote the set relative coordinates,

$$x = \begin{pmatrix} x_1 \\ x_2 \end{pmatrix}. \tag{7}$$

Here we choose the Jacobi coordinates:

$$x_1 = r_1 - r_2, \tag{8}$$

$$x_2 = \frac{m_1 r_1 + m_2 r_2}{m_1 + m_2} - r_3, \tag{9}$$

where $r_i$ ($i = 1, 2, 3$) is the single-particle coordinate of the $i$th particle. The three-body wave function $|\Psi^{(3)}\rangle$ is expanded on a basis of correlated Gaussians (CG) [13],

$$|\Psi^{(3)}\rangle = \sum_k c_k \mathcal{S} \exp\left(-\frac{1}{2}\tilde{x} A_k x\right), \tag{10}$$

where $\mathcal{S}$ is symmetrizer acting on the three particles (Model I) or on the $\{1, 2\}$ subset (Model II), and $A_k$ is the positive-definite 2×2 symmetric real matrix which characterizes the $k$th CG. The energy and the expansion coefficients $\{c_k\}$ are determined by solving a generalized

eigenvalue problem. To optimize the non-linear variational parameters entering the $A_k$, we employ the stochastic variational method [13, 14]. Since we have to treat simultaneously two different scales, atomic and nuclear, we adopt the following strategy in the search for the variational parameters. Suppose that we have already a basis of $K$ CG: A number of candidates for the additional $A_{K+1}$ matrices are generated randomly with their elements either at the nuclear or atomic scale. For small $K$, we select the matrix providing the minimum energy. Once the energy is converged up to a certain number of digits, the additional CG are generated only with elements at the nuclear scale. This procedure is efficient, particularly with large $\lambda$, where the wave function changes drastically at short distances. In our calculations, we have increased the size of the basis until the energy is converged within $10^{-4}$.

# 4 Factorization of the long- and short-range contributions

## 4.1 Deser-Trueman formula

The energy shift of two-body exotic atoms is often estimated with the Deser-Trueman (DT) formula [15, 16].

$$\delta E^{(2)} = \frac{2\pi}{\mu} |\Psi_0^{(2)}(0)|^2 a, \tag{11}$$

where $\mu$ is the reduced mass, $\Psi_0^{(2)}(0)$ is the relative wave function at the origin obtained with the long-range potential alone, and $a$ is the scattering length calculated with by the short-range potential alone. Note the remarkable factorization of the long-range and short-range contributions in the DT formula. Fig. 2 shows a comparison of the ground-state energy of two identical bosons interacting with Eqs. (2) and (3), calculated either exactly or by the DT formula, with the strength $\lambda$ of the short-range potential varied continuously.

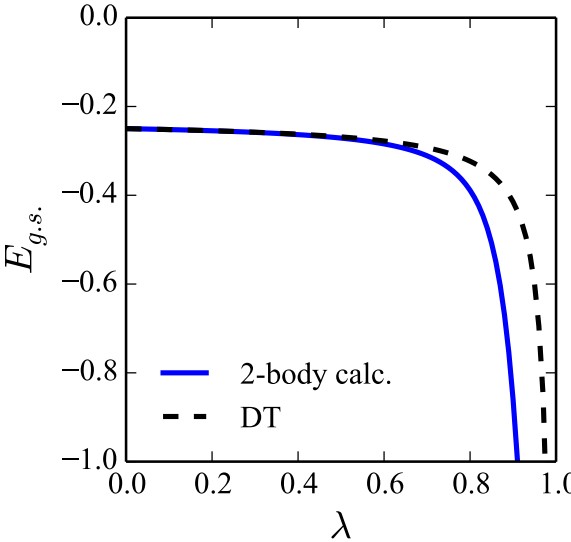

Figure 2: Comparison of the two-body ground-state energy, $E_{g.s.}$, obtained with the full two-body calculation and the DT formula.

At small $\lambda$, the DT formula reproduces well the ground-state energy obtained by direct two-body calculations. However, the DT formula deviates from the two-body calculation as $\lambda$ increases. This shows that the energy shift involves higher-order corrections, beyond the

simple scattering length in the DT formula. A parallel question is whether or not the energy shift can still be factorized into the long- and short-range contributions in large $\lambda$ region. For more quantitative discussion, we introduce in the next subsection the determinant method.

## 4.2 Determinant Method

To evaluate quantitatively the validity of the factorization of the energy shift, we use the following method. Let $M$ be the matrix of the energy-shifts for a series of discretized strengths $\lambda_1, \lambda_2, \ldots$ and several long-range potentials, as spelled out in Table 1.

Table 1: Matrix $M$ constructed from the energy shifts obtained with different long- and short-range potentials.

|  | $\lambda_1$ | $\lambda_2$ | $\lambda_3$ | $\cdots$ |
|---|---|---|---|---|
| $LR_\mathrm{I}$ | $\delta E(LR_\mathrm{I}, \lambda_1)$ | $\delta E(LR_\mathrm{I}, \lambda_2)$ | $\delta E(LR_\mathrm{I}, \lambda_3)$ | $\cdots$ |
| $LR_\mathrm{II}$ | $\delta E(LR_\mathrm{II}, \lambda_1)$ | $\delta E(LR_\mathrm{II}, \lambda_2)$ | $\delta E(LR_\mathrm{II}, \lambda_3)$ | $\cdots$ |
| $LR_\mathrm{III}$ | $\delta E(LR_\mathrm{III}, \lambda_1)$ | $\delta E(LR_\mathrm{III}, \lambda_2)$ | $\delta E(LR_\mathrm{III}, \lambda_3)$ | $\cdots$ |
| $\vdots$ | $\vdots$ | $\vdots$ | $\vdots$ | $\ddots$ |

If the level shift can be factorized as the product of a contribution from the long-range potential and another from the short-range part potential, as in the DT formula, the determinant of any $2 \times 2$ submatrix $S_2$ taken from $M$ must be zero. For example, for $\lambda_i$ when $\delta E(LR, \lambda_i)$ is the product of separated contributions from the long-range and short-range interactions, that is $\delta E(LR, \lambda_i) = A_{LR}B_{SR}(\lambda_i)$. The submatrix $S_2$ is defined by

$$S_2 = \begin{pmatrix} \delta E(LR_\mathrm{I}, \lambda_1) & \delta E(LR_\mathrm{I}, \lambda_2) \\ \delta E(LR_\mathrm{II}, \lambda_1) & \delta E(LR_\mathrm{II}, \lambda_2) \end{pmatrix} = \begin{pmatrix} A_{LR_\mathrm{I}}B_{SR}(\lambda_i) & A_{LR_\mathrm{I}}B_{SR}(\lambda_{i+1}) \\ A_{LR_\mathrm{II}}B_{SR}(\lambda_i) & A_{LR_\mathrm{II}}B_{SR}(\lambda_{i+1}) \end{pmatrix}. \tag{12}$$

Considering the determinant of $S_2$, it can be easily proven that the $\det S_2$ is zero analytically as

$$\det S_2 = A_{LR_\mathrm{I}}B_{SR}(\lambda_i)A_{LR_\mathrm{II}}B_{SR}(\lambda_{i+1}) - A_{LR_\mathrm{I}}B_{SR}(\lambda_{i+1})A_{LR_\mathrm{II}}B_{SR}(\lambda_i) = 0. \tag{13}$$

On contrary, when the energy shift is not separable, then $\det S_2$ is not necessarily zero. Practically, to get the variation of the potentials, we take the two long-range potentials with $\mu_{LR} = 10$ and $\mu_{LR} = 30$ and different $\lambda$s at intervals of 0.01 ($\lambda_{i+1} - \lambda_i = 0.01$).

## 4.3 Factorization of long- and short-range contributions in two-body system

Let us show how the determinant method works for the two-body system. Figure 3 plots $|\det S_2|$ as a function of the strength of the short-range potential $\lambda$. To appreciate what $\det S_2 \simeq 0$ means, we take into account the order of magnitude of the elements of $S_2$ and the accuracy of the calculation. In Fig. 2b, this corresponds to $|\det S_2| \lesssim 10^{-6}$. The shaded area in Fig. 3 indicates the possible regions where the numerical error dominates Here the range of the strength $\lambda$ where the factorization holds is seen to be about $\lambda \lesssim \lambda_c = 0.6$. Interestingly this is the range of $\lambda$ for which the DT formula works very well.

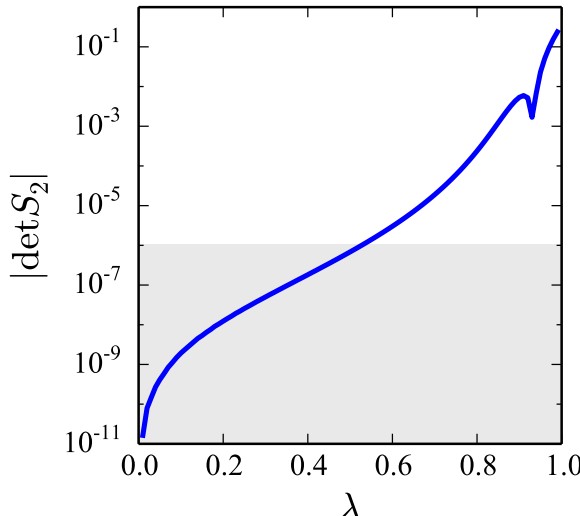

Figure 3: The determinant of the $S_2$ of the two-body system. The shaded region shows the region that in which $\det S_2$ can be considered to be zero.

## 5  Discussions: Three-body correlations

To discuss the results of three-body models, the two-body DT formula is extended to the three-body case as [12]

$$\sum_{i>j=1}^{3} \frac{2\pi}{\mu_{ij}} |\Psi_{0,ij}^{(3)}(0)|^2 a_{ij}, \tag{14}$$

where $\mu_{ij}$, $a_{ij}$ are respectively the reduced mass and the scattering length obtained only with the short-range potential of the $i$th and $j$th particles, and $|\Psi_{0,ij}^{(3)}(0)|^2$ is defined by

$$|\Psi_{0,ij}^{(3)}(0)|^2 = \frac{\langle \Psi_0^{(3)} | \delta(\boldsymbol{r}_i - \boldsymbol{r}_j) | \Psi_0^{(3)} \rangle}{\langle \Psi_0^{(3)} | \Psi_0^{(3)} \rangle}. \tag{15}$$

$\Psi_0^{(3)}$ is the wave function obtained only with the long-range potential. Note that the extended DT formula keep the form of a sum of products of contributions from the long- and short-range potentials.

The upper panel of Fig. 4a shows comparison between the ground-state energy obtained by the full three-body calculation and the extended DT formula of Eq. (15) for Model I. At small values of $\lambda$, the energy shift is small and shows a flat behavior. The extended DT formula reproduces well the energy shift of the three-body calculation in this flat region. From the lower left panel, one can see that the factorization is also valid in that region. Then the energy shift drops rapidly at some $\lambda$, and the factorization breaks down simultaneously (we again estimated the area for which a vanishing of the determinant makes sense, given the order of magnitude of the matrix elements and the accuracy of the calculation). A departure for the DT approximation is observed at about $\lambda_c \simeq 0.4$, while in the two-body case, a similar departure occurred only at $\lambda_c \simeq 0.6$. This is because in the latter case, a purely nuclear state requires $\lambda = 1$, for which $a \to \infty$, while in the former case, a Borromean three-body bound state occurs for $\lambda \simeq 0.8$. Hence the atomic spectrum is "pulled down" earlier.

Figure 4b displays the same plots for Model II. The energy shift and the determinant exhibit the same qualitative behavior but the critical strength becomes much larger, $\lambda_c \simeq 0.8$. This is

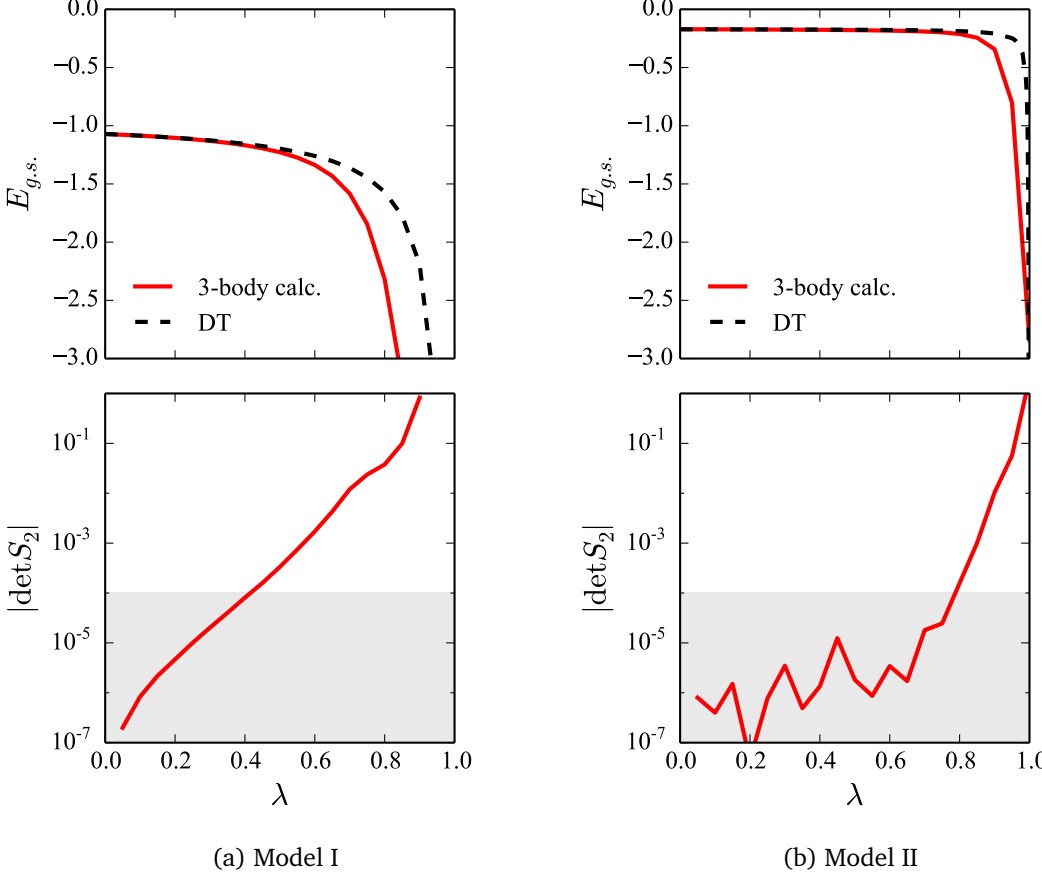

Figure 4: (Upper) Ground-state energy $E_{g.s.}$ obtained by the full three-body calculation and the DT formula. (Lower) $|\det S_2|$ values calculated with the three-body calculations.

because of the repulsive long-range potential between two identical bosons, which suppresses the three-body correlations. The level shift of such a three-body system is determined only by the pairwise correlations and is of factorizable form.

## 6 Summary

Accurate three-body calculations have been performed to evaluate three-body correlations in exotic-atom-like three-body systems. The interaction, which is pairwise, consists of a Coulomb-type of long-range interaction and a short-range potential whose strength is varied. Two models have been considered. Model I consists of three identical bosons. Model II includes two identical bosons of mass $m_{1,2} = 1$ and a third particle of mass $m_3 = 1/2$, and opposite charge. The factorization property of the long- and short-range contributions to the energy shift have been examined quantitatively by the determinant method.

We find that, when the strength of the nuclear interaction is increased, the factorization and the dominance of two-body correlations break down earlier when the same long- and short-range potentials are applied to all pairs (Model I), whereas the three-body correlations are much smaller with Model II in which only two pairs interact. This is intimately related to the early or delayed occurrence of a Borromean three-body bound state in the nuclear potential.

As a further extension of this study, the analysis of the excited states is underway for a

general understanding of the many-body correlations and of the level rearrangement. In particular, we shall extend the method of the determinant to larger submatrices to probe whether the energy shift is a sum of products of long- and short-range terms, rather than a mere product. We also aim at investigating such exotic-atom-like systems with a complex potential to take the meson-baryon absorption effect into account. This is, indeed, an important aspect of the $\bar{K}N$ interaction [4, 8].

## Acknowledgements

We acknowledge the collaborative research program 2019, information initiative center, Hokkaido University.

**Funding information** This work was in part supported by JSPS KAKENHI Grants No. 18K03635, No. 18H04569, and No. 19H05140.

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
