# Peer review of "Three-body correlations in mesonic-atom-like systems"

_SciPost Physics Proceedings, doi:SciPost Phys. Proc. 3, 051 (2020)_

## Round 1 · Referee Report · Anonymous (Referee 1) · 2019-11-17

Report
The authors study a mesonic atom system, with their accurate three-body calculation method. They have found that the calculated 3-body energy is in good agreement with the 2-body correlation model, DT model they call, for a weakly interacting regime, while the disagreement becomes significant for stronger interaction, in particular close to the unitary regime. Although their model is too simple to directly apply to the mesonic atom system they aim to study, their results have clear physical insights which may be useful for their forthcoming studies which should be more realistic. So I think the manuscript is ready for publication as it is.

---

## Editorial Decision

published